# CXCR3 Inhibition Blocks the NF-κB Signaling Pathway by Elevating Autophagy to Ameliorate Lipopolysaccharide-Induced Intestinal Dysfunction in Mice

**DOI:** 10.3390/cells12010182

**Published:** 2023-01-01

**Authors:** Cheng Zhang, Yian Deng, Yingsi Zhang, Tongtong Ba, Sai Niu, Yiqin Chen, Yuan Gao, Hanchuan Dai

**Affiliations:** College of Veterinary Medicine, Huazhong Agricultural University, No. 1 Shizishan Street, Wuhan 430070, China

**Keywords:** CXCR3, autophagy, NF-κB signaling pathway, inflammation, intestinal epithelial cells

## Abstract

Autophagy is a cellular catabolic process in the evolutionarily conservative turnover of intracellular substances in eukaryotes, which is involved in both immune homeostasis and injury repairment. CXCR3 is an interferon-induced chemokine receptor that participates in immune regulation and inflammatory responses. However, CXCR3 regulating intestine injury via autophagy along with the precise underlying mechanism have yet to be elucidated. In the current study, we employed an LPS-induced inflammatory mouse model and confirmed that CXCR3 knockout significantly attenuates intestinal mucosal structural damage and increases tight junction protein expression. CXCR3 knockout alleviated the LPS-induced increase in the expression of inflammatory factors including TNF-α, IL-6, p-65, and JNK-1 and enhanced autophagy by elevating LC3II, ATG12, and PINK1/Parkin expression. Mechanistically, the function of CXCR3 regarding autophagy and immunity was investigated in IPEC-J2 cells. CXCR3 inhibition by AMG487 enhanced autophagy and reduced the inflammatory response, as well as blocked the NF-κB signaling pathway and elevated the expression of the tight junction protein marker Claudin-1. Correspondingly, these effects were abolished by autophagy inhibition with the selective blocker, 3-MA. Moreover, the immunofluorescence assay results further demonstrated that CXCR3 inhibition-mediated autophagy blocked p65 nuclear translocation, and the majority of Claudin-1 was located at the tight junctions. In conclusion, CXCR3 inhibition reversed LPS-induced intestinal barrier damage and alleviated the NF-κB signaling pathway via enhancing autophagy. These data provided a theoretical basis for elucidating the immunoregulatory mechanism by targeting CXCR3 to prevent intestinal dysfunction.

## 1. Introduction

Intestinal epithelial cells play key roles in the digestion and absorption of nutrients, as well as in preventing pathogenic bacterial invasion, immune regulation, and maintaining intestinal homeostasis [1,2]. The tight junctions of the intestinal epithelium maintain cell connections and provide a physical barrier against luminal inflammatory molecules related to inflammation and infection [3]. The intestinal mucosal barrier is regulated in response to physiological and immunological stimuli, and it is crucial for immune homeostasis [4]. The impaired integrity and structure of the tight junctions barrier induced by proinflammatory cytokines, pathogenic bacteria, lipopolysaccharides (LPS), and pathological conditions usually result in a forcible activation of the immune system and chronic inflammation in different tissues [5] and are increasingly implicated in controlling disease development [6]. Therefore, protecting the integrity and homeostasis of the tight junction barrier is an effective way to prevent inflammatory bowel disease (IBD).

Autophagy is a fundamental biological process which maintains intracellular metabolic homeostasis and contributes to cytoplasmic quality control, cellular metabolism, and immunity [7,8]. Studies have elucidated the involvement of autophagy in the maintenance of the intestinal epithelial barrier [9,10,11,12], and autophagy is involved in both homeostasis and intestine repair, supporting the intestinal barrier function in response to cellular stress through tight junction regulation [13]. It is shown that restoring autophagy can reduce the NF-κB signaling pathway to limit epithelial inflammatory responses and oxidative stress injury [14,15,16]. Moreover, intestinal epithelial cells’ deficiency in autophagy genes leads to compromised protection against enteric infections [13,17,18]. Autophagy likely underscores the protection and recovery from infection and inflammation and seems to be a new therapeutic target and diagnostic marker for intestinal inflammation.

CXCR3—also called G-protein-coupled receptor 9 or CD183—is an interferon-induced chemokine receptor that can be expressed on NK T cells, monocytes, CD8^+^ T cells, Th1 T cells, NK cells, dendritic cells, and cancer cells [19,20]. There are three isoforms in humans, including CXCR3A, CXCR3B, and CXCR3-alt, while mice have a single isoform of the CXCR3 receptor, namely, CXCR3A [21]. The chemokine/chemokine receptor system is widely involved in immune cell recruitment and inflammation [22]. CXCR3 can react with CXCL4, CXCL9 (MIG), CXCL10 (IP-10), and CXCL11 (I-TAC/IP-9) to induce effector cells to localize to inflammation sites, which is involved in regulating the migration, activation, and differentiation of immune cells, as well as affecting immunity, inflammation, and angiogenesis [23,24]. Meanwhile, CXCR3 activation contributes to autophagy suppression and the concomitant effects of antiretroviral therapy [25], and this implies that CXCR3 activation is the missing link between infection and autophagy impairment associated with synaptic injury and neuronal loss [25]. Moreover, CXCR3 plays a pivotal role in inducing the production of cytokines, macrophage infiltration, and causing autophagy deficiency and ER stress in NASH development [26]. CXCR3 is strongly overexpressed in the intestinal mucosa of IBD patients and celiac disease (CD) patients and contributes to recruiting proinflammatory cells to the colon during colitis [27,28]. However, whether CXCR3 regulates inflammation and repairs injury though autophagy remains uncertain in the intestine.

CXCR3 is demonstrated to be of great significance to immunity and inflammation [29]. Yet, the function of CXCR3 regarding intestinal epithelial cells’ autophagy and inflammation has yet to be elucidated. In this study, the relationship between autophagy and inflammation was investigated in vitro and in vivo using CXCR3^−/−^ mice and porcine intestinal epithelial cells (IPEC-J2). The results could contribute to understanding the mechanisms in the recovery of intestinal function during the inflammation process by targeting CXCR3.

## 2. Materials and Methods

### 2.1. Animal Handling and Sample Collection

Female C57BL/6J (Wild-type, WT) mice (6 weeks) and female CXCR3 knockout (CXCR3^−/−^) C57BL/6J mice (6 weeks) were purchased from the Experimental Animal Center (Huazhong Agricultural University, China) according to the Institutional and National Institutes of Health guidelines for humane animal use. The animal certificate number was SYXK (Hubei) 2020-0084. This work was approved by Huazhong Agricultural University’s Animal Experimental Ethics Committee (HZAUMO-2020-0078). All mice were housed in the individual ventilated cages and maintained in a temperature- and humidity-controlled room (22 °C, 50% relative humidity) with 12 h light/dark cycles. The mice were randomly divided into four groups (n = 8) as follows: the WT group, WT + LPS group, CXCR3^−/−^ group, and CXCR3^−/−^ + LPS group. The mice were intraperitoneally injected with LPS (10 mg/kg) or the same dose of normal saline [30]. After an interval of 24 h, the mice were euthanized by an intraperitoneal injection of sodium pentobarbital (30 mg/kg) and sacrificed. The jejunum tissue was collected for future analysis.

### 2.2. Cell Culture

IPEC-J2 cells were seeded in DMEM/F12 medium (Gibco, New York, USA), which contains 10% fetal bovine serum (Gibco, New York, USA) and 1% penicillin–streptomycin solution, and then incubated in a constant temperature incubator (Thermo Fisher Scientific, Waltham, MA, UAS) at 37 °C and 5% CO_2_. When the cell abundance reached 70–80%, the cells were seeded in a 6-well plate or a 12-well plate. The cells were divided into four groups, namely, the Control group, LPS (10 μg/mL) group (Sigma, St. Louis, MO, USA), LPS + AMG487 (1 μM) (MedChemExpress, Shanghai, China) group, and LPS + AMG487 + 3-MA (5 mM) (MedChemExpress, Shanghai, China) group [31,32,33,34]. The cells were treated for 12 h, and then the samples were collected for further analysis.

### 2.3. HE Staining

The jejunum tissue was fixed with formalin, dehydrated by ethanol, and embedded in paraffin. The paraffin-embedded jejunum was adjusted to a thickness of 4 μm and then analyzed by hematoxylin and eosin staining (H&E). The stained tissue sections were observed with an optical microscope (Olympus BX51, Tokyo, Japan), and pictures were taken for preservation. The intestinal villi height and the crypts depth of the tissue were measured.

### 2.4. RNA Extraction and Real-Time Fluorescence Quantification (qPCR)

Total RNAs were extracted from the jejunum tissue with Trizol reagent (Invitrogen, Carlsbad, USA), and 1 μg RNA was reversed by using Hifair^®^II 1st Strand cDNA Synthesis SuperMix for qPCR. Super mix (Yeasen, Shanghai, China) was used for qPCR in a Roche LightCycler 480 PCR system (Roche, Basel, Swiss). The 2^−ΔΔCT^ method was used to measure the abundance of mRNA transcripts and was normalized to GAPDH mRNA. All samples are at least three replicates. The primers (Qingke, Wuhan, China) used are listed in Table 1.

### 2.5. Western Blot

Proteins were extracted from IPEC-J2 cells and jejunum tissues using the RIPA lysis buffer (Beyotime, Beijing, China) containing 1 mM PMSF (Beyotime, Beijing, China) and 1 mM phosphatase inhibitor (Beyotime, Beijing, China). For nuclear protein preparations, the cells were washed twice with PBS, and the proteins in the nucleus were extracted using a Nuclear and Cytoplasmic Protein Extraction Kit (Beyotime, Beijing, China), following the manufacturer’s protocol. The protein concentration was determined by a bicinchoninic acid (BCA) determination kit (Beyotime, Beijing, China), separated by 12% polyacrylamide gel electrophoresis containing 0.1% SDS, and transferred to a PVDF membrane. After blocking with the solution containing 20 mM Tris-HCl, 0.1% Tween, and 5% nonfat dry milk at room temperature for 2 h, the membranes were incubated with the primary antibody overnight at 4 °C. Primary antibodies including PINK1, Parkin, ATG12, LC3, and p65 were purchased from Proteintech (Wuhan, China). JNK1, p-JNK1, TNF-α, and IL-6 were purchased from Wanleibio (Shengyan, China). p-p65 and Claudin-1 were purchased from ABclonal (Beijing, China). GAPDH and Histone H3 were purchased from Servicebio (Wuhan, China), and p62 was purchased from Cell Signaling Technology (Dallas, TX, USA). The primary antibodies were removed by washing three times. The secondary antibodies (Servicebio, Wuhan, China) were incubated at room temperature for 90 min and then washed three times with TBST. GAPDH and histone H3 were used as internal controls. An electrochemiluminescence (ECL) imaging system (Biotanon-5200, Shanghai, China) was used to scan the immunoreactivity bands of the target proteins, and Image J software (National Institutes of Health, Bethesda, Rockville, MD, USA) was used to quantify the digital data of the integrated optical density.

### 2.6. Indirect Immunofluorescence

IPEC-J2 cells were cultured in a 12-well plate. When the cell density reached 70%, the cells were co-treated with or without LPS, AMG487, and 3-MA. After 12 h, the cells were collected and fixed with 4% paraformaldehyde at 37 °C for 20 min. Next, the cells were incubated with PBS containing 1% bovine serum albumin (BSA) and 1% Triton X-100 at 37 °C for 2 h and then with a 1:100 dilution of Claudin-1 (ABclonal, Wuhan, China), p65 (Proteintech, Wuhan, China), and LC3 (Proteintech, Wuhan, China) antibodies overnight at 4 °C. After washing three times, 30 min later, the cells were conjugated with Alexa Fluor 488 affinity goat anti-rabbit IgG (H + L) (Proteintech, Wuhan, China) at room temperature for 1 h, and, finally, a laser confocal microscopy system (LSM880, Zeiss, Germany) was used to observe the protein expression in the cells.

### 2.7. Statistical Analysis

The data are expressed as the mean ± SEM of at least three independent experiments for each cellular experimental group. Statistical analysis was performed on the group by using one-way analysis of variance (ANOVA) followed by Tukey’s post hoc test in groups of more than two, which were used to establish statistical significance with GraphPad Prism version 9.0 (* *p* < 0.05; ** *p* < 0.01; *** *p* < 0.001; ns indicates no significance).

## 3. Results

### 3.1. CXCR3 Knockout Attenuates Intestinal Mucosal Structural Damage and Increases Tight Junction Protein Expression in the Mouse Intestine

CXCR3 is a kind of chemokine with the characteristic of recruiting the inflammatory microenvironment and enhancing the progression of murine tissue lesions [35]. However, the function of CXCR3 in intestinal mucosal structural damage needs to be further clarified. In the current study, we first confirmed that CXCR3^−/−^ mice cannot express the CXCR3 protein (Appendix A). Subsequently, C57BL/6J WT mice and CXCR3^−/−^ mice were intraperitoneally injected with LPS 10 mg/kg. After 24 h, the mice were sacrificed, and samples were collected (Figure 1A). The morphology and the tight junction protein expression of the intestine were explored. HE staining results showed that LPS administration destroyed the intestine’s structural integrity. The part of the intestinal villi was damaged and shed (Figure 1B,C). The crypt depth and the mucosal thickness of the intestine in the LPS group were significantly higher than those in the WT group (Figure 1B,D). Moreover, the boundary between the mucosa and muscularis mucosa was irregular (Figure 1B). In contrast, the intestine’s morphological structure was observed to be intact in the CXCR3^−/−^ group (Figure 1B–D). CXCR3 knockout alleviated LPS-induced intestinal villi shortening and reversed the mice intestine histologic changes caused by LPS administration (Figure 1B–D). In addition, the expression of the tight junction protein was explored. LPS treatment could reduce the expression of the Claudin-1 protein. However, the expression of the Claudin-1 protein was elevated in CXCR3^−/−^ mice, and the decrease was reversed in CXCR3^−/−^ + LPS mice (Figure 1E,F). The results showed that the knockout of CXCR3 attenuated LPS-induced intestinal mucosal structural damage and increased tight junction protein expression, which might contribute to maintaining intestinal structure and function.

### 3.2. CXCR3 Knockout Promotes Autophagy in the Mouse Intestine

As a highly conserved degradation and regeneration program in cells, autophagy acts as an intracellular garbage disposal station, which can eliminate pathogens, promote development, inhibit tumors, maintain cell homeostasis, and participate in immune responses [12,36]. To investigate the relationship between CXCR3 and autophagy in the mice intestine, the mRNA and protein expression levels of autophagy-related genes were analyzed. As shown in Figure 2, LPS stimulation and CXCR3 knockout can induce the mRNA and protein upregulation of autophagy-related genes, as confirmed by the increased expression of ATG12 (Figure 2A) and the LC3II/LC3I ratio (Figure 2B), as well as the upregulation of the mitophagy markers PINK1 (Figure 2C) and Parkin (Figure 2D). However, for SQSTM1/p62, a prototype autophagy receptor and an autophagic substrate, the expression was blocked in the LPS group and CXCR3^−/−^ group (Figure 2E). Importantly, CXCR3 knockout can further increase the mRNA and protein expressions of ATG12, LC3 II, PINK1, and Parkin (Figure 2A–D) and significantly decrease the expression of p62 mRNA and the protein when CXCR3^−/−^ mice are stimulated with LPS (Figure 2E). The results showed that CXCR3 knockout can promote LPS-induced intestinal autophagy in mice intestine tissue.

### 3.3. CXCR3 Knockout Blocks the Activation of the NF-κB Signaling Pathway in the Mouse Intestine

The NF-κB signaling pathway has been recognized as a critical regulator of immunity and inflammation [37,38]. To examine the effect of CXCR3 on the NF-κB signaling pathway in the mouse intestine, the expression levels of NF-κB signaling pathway-related genes in WT and CXCR3^−/−^ mice were explored. As shown in the results, the mRNA and protein expressions of IL-6 (Figure 3A), TNF-α (Figure 3B), p65 (Total) (Figure 3C), JNK1 (Figure 3D), and JNK1 phosphorylation (Figure 3E) were elevated following LPS treatment. However, compared with the control group, the changes in proinflammatory factors expression were not significant in CXCR3^−/−^ mice jejunum (Figure 3). Notably, the knockout of CXCR3 reversed the mRNA and protein expressions of the proinflammatory factors TNF-α and IL-6 after being challenged with LPS (Figure 3A,B). We further found that the expression of total p65, JNK1, and JNK1 phosphorylation was also blocked in jejunal (Figure 3C,D). These results implied that CXCR3 knockout can exert an anti-inflammatory property by inhibiting the activation of the NF-κB and JNK signaling pathways in the mice intestine tissue.

### 3.4. CXCR3 Inhibition Promotes Autophagy in IPEC-J2 Cells

CXCR3 knockout was confirmed to inhibit inflammation in mice intestine tissue. To further verify the essential role of CXCR3 in autophagy in IPEC-J2 cells, the CXCR3 antagonist AMG487 and the autophagy inhibitor 3-MA were used to investigate the effect of CXCR3 on autophagy in IPEC-J2 cells. The results demonstrated that CXCR3 expression was markedly inhibited by AMG487 (Appendix A). LPS treatment could significantly induce the expression of the Parkin (Figure 4A,B), PINK1 (Figure 4A,C), LC3 II (Figure 4A,D), and ATG2 (Figure 4A,E) proteins and reduce the expression of the p62 protein (Figure 4A,F). Moreover, CXCR3 inhibition by AMG487 could further increase the expression of the above-mentioned protein and significantly decrease p62 protein expression (Figure 4A–F). However, autophagy inhibition by 3-MA could obviously reverse these effects (Figure 4A–F). Additionally, indirect immunofluorescence assays were adopted to explore the autophagy levels and indicated that the distribution of LC3 was diffuse in the cytoplasm in LPS-treated cells, and CXCR3 inhibition by AMG487 could further elevate the accumulation of LC3 around the nucleus (Figure 4G). However, autophagy suppression by 3-MA resulted in the distribution of LC3 in the cytoplasm and was significantly reduced (Figure 4G). Collectively, these results demonstrated that CXCR3 inhibition can promote autophagy in IPEC-J2 cells.

### 3.5. CXCR3 Inhibition Blocks the NF-κB Signaling Pathway by Promoting Autophagy in IPEC-J2 Cells

Recent developments revealed a crucial role for the autophagy pathway and proteins in immunity, which control inflammation through regulatory interactions with innate immune signaling pathways [8,39,40]. To explore the effect of CXCR3-mediated autophagy on innate immune-related proteins, the NF-κB signaling pathway was investigated in IPEC-J2 cells. The results indicated that CXCR3 inhibition by AMG487 could deteriorate the expression of TNF-α (Figure 5A,B), IL-6 (Figure 5A,C), p65 phosphorylation (Figure 5A,D), and JNK1 (Figure 5A,E) induced by LPS. CXCR3 inhibition suppressed p65 phosphorylation and JNK1 phosphorylation (Figure 5A). However, p-JNK1/JNK1 changes showed no significant difference (Figure 5A,F). Additionally, compared with the LPS + AMG487 group, autophagy inhibition by 3-MA significantly increased TNF-α and IL-6 expression, which was accompanied by p65 phosphorylation elevation (Figure 5). Taken together, these results showed that CXCR3 inhibition can block the NF-κB signaling pathway, which might be dependent on autophagy.

### 3.6. CXCR3 Inhibition Blocked p65 Nuclear Translocation by Promoting Autophagy in IPEC-J2 Cells

To further explore the effect of CXCR3 on the NF-κB signaling pathway by promoting autophagy, p65 expression in the cytoplasm and in the nucleus and its nuclear localization were investigated. Western Blot assays showed that LPS could induce the expression of p65 in the nucleus, which can be restrained by CXCR3 inhibition. In contrast. autophagy inhibition increased the expression of p65 in the nucleus compared with the LPS-AMG487 group (Figure 6A,B). However, the p65 level showed no significant changes in the cytoplasm (Figure 6C,D). Additionally, the CXCR3-mediated autophagy-induced nuclear translocation of p65 was observed using laser confocal microscopy. An indirect immunofluorescence assay demonstrated that LPS could induce p65 accumulation diffusely distributed in the nucleus. CXCR3 inhibition resulted in a decrease in p65 expression in the nucleus. However, autophagy suppression contributed to p65 accumulation that was diffusely distributed in the nucleus (Figure 6C,D). Collectively, these results suggested that CXCR3 inhibition can inhibit the nuclear translocation of p65 by promoting autophagy.

### 3.7. CXCR3 Inhibition Promotes Tight Junction Protein Expression in IPEC-J2 Cells

Inflammation caused by either intrinsic or extrinsic toxins results in intestinal barrier dysfunction, and defective tight junctions can induce intestinal epithelial damage [41,42]. We then explored the tight junction protein expression in IPEC-J2 cells. The Western Blot assay demonstrated that CXCR3 inhibition alleviated the LPS-induced decrease in the expression of the tight junction protein marker Claudin-1 and can be elevated by autophagy inhibition (Figure 7A,B). Mechanistically, immunofluorescence results demonstrated that the majority of Claudin-1 is located at the tight junctions, as we expected. LPS reduced the expression of the Claudin-1 protein, and CXCR3 inhibition reversed the LPS-reduced expression of the tight junction protein. Moreover, autophagy inhibition can decrease Claudin-1 expression, which is consistent with the results of the Western Blot (Figure 7C). In conclusion, CXCR3 inhibition can reverse LPS-induced intestinal barrier damage and improve the intestinal expression of tight junction proteins via enhancing autophagy.

## 4. Discussion

The intestinal tract has irreplaceable functions in the body’s healthy development, such as the signal regulation of immunity, the digestion and absorption of nutrients, material conversion, the recognition response of foreign microorganisms, and simultaneously defense against pathogen invasion [43,44]. The intestinal mucosal barrier is the first line of defense against the permeation of luminal contents and performs numerous biological functions [45,46,47]. The intestine is extremely sensitive to pressure. Pathogens, intestinal microbes, and environments exert unavoidable pressure on the intestinal barrier, which causes diarrhea, IBD, intestinal irritable syndrome (IBS), and other diseases [48].

The integrity of the intestinal mucosal barrier is heavily dependent on the tight junctions. One of the most important roles of the tight junctions is to provide a physical barrier to luminal inflammatory molecules. Tight junction proteins such as occluding, claudin-1, and zo-1 are regarded as the targets and effectors of immune homeostasis and are internalized through an NF-kB-dependent pathway [49]. The reduction of NF-kB activation may repair defects in epithelial barrier function, enhance tight junction expression, and reduce diarrhea [50,51]. JNK1 is a subfamily of (MAPK) mitogen-activated protein kinases and is also involved in the regulation of intestinal inflammatory cytokines production [52,53]. The tight junctions are disrupted, and large amounts of proinflammatory cytokines are produced, resulting in immune dysregulation in the inflamed tissue of patients with ulcerative colitis [54]. The impaired integrity and structure of the tight junction barrier result in a forcible activation of immune cells and inflammation in different tissues [5]. Conversely, the modulation of tight junction barrier conductance, especially within the gastrointestinal tract, can impact immune homeostasis and diverse pathologies [49]. In the current research, we reported the striking impact of CXCR3 inhibition on the intestinal morphology and tight junction expression barrier in CXCR3^−/−^ mice. CXCR3 knockout and inhibition significantly decreased inflammatory cytokine expression evoked by LPS in mice and IPEC-J2 cells. The NF-KB pathway and JNK1 signaling were alleviated in vitro and in vivo, which indicated that CXCR3 inhibition blocked inflammation in the intestine. These data suggested that CXCR3 inhibition enhanced barrier integrity, elevated tight junction expression, and held out the protective or therapeutic efficiency of CXCR3 for inflammatory bowel disease.

Multidirectional studies have shown that autophagy is an important factor in regulating intestinal mucosal damage and recovery and contributes to regulating the secretion of antimicrobial peptides, removing damaged organelles, eliminating bacteria, and regulating enteritis [13,55]. The tight link between autophagy and intestinal physiology has been illustrated by the observation of the strong association between genes of the autophagy pathway [13,36]. It is shown that autophagy can enhance the tight junction barrier function and significantly increase the TER in intestinal epithelial cells [56]. SIRT1/PGC-1 pathway activation triggers autophagy/mitophagy and attenuates oxidative damage in intestinal epithelial cells, which might be a protective mechanism for increasing tight junction integrity against oxidative stress-mediated ROS production in IPEC-1 cells [12]. However, the dysregulation of the autophagy process causes the disruption of several aspects of the intestinal epithelium, which subsequently leads to inappropriate immune responses, inflammation, and intestinal injury [57,58]. In this article, the autophagy-related proteins were explored in CXCR3^−/−^ mice as well as in IPEC-J2 cells. The decreased level of p62, together with the transformation from LC3B-I to LC3B-II, was noticed, and the expression of autophagy markers including PINK1, Parkin, and ATG12 was also elevated in CXCR3^−/−^ mice. These changes have been observed in IPEC-J2 cells following CXCR3 antagonist AMG487 treatment. The data indicated that CXCR3 knockout and inhibition can elevate autophagy in vitro and in vivo, suggesting that CXCR3-mediated autophagy contributes to intestinal homeostasis under an inflammation status.

Previous studies have shown that CXCR3 can inhibit autophagy and promote inflammation [25,26]. Farrell GC determined the key role of CXCR3 in the regulation of the autophagosome-lysosome system, and knocking out CXCR3 can inhibit polyubiquitinated proteins and ER stress/UPR in steatohepatitis, which is conducive to the recovery of autophagy [26,59,60]. CXCR3 can damage liver cell autophagy by inducing lysosomal damage and endoplasmic reticulum stress, and it has an important regulatory role in the development of nonalcoholic steatohepatitis [61]. Neurons exposed to the chemokine CXCL12 can activate the expression of CXCR3, inhibit neuronal autophagy, reduce neuronal activity, and cause neuroinflammation [25]. Becn1 deficiency leads to the stabilization of MEKK3 in neutrophils and the abnormal activation of p38, which is mediated by CXCL9/CXCR3 chemotaxis. Moreover, AMG487, a specific inhibitor of the chemokine receptor CXCR3, is the targeted blocker of CXCR3 and improves inflammatory symptoms by blocking the inflammatory cycle [62,63]. These results indicate that CXCR3 plays an important role in the regulation of inflammatory diseases by mediating autophagy. In the present study, the NF-κB signaling pathway and tight junction expression were investigated in IPEC-J2 cells treated with AMG487 and 3-MA. CXCR3 inhibition by AMG487 can further enhance autophagy levels. In addition, the NF-κB signaling pathway was blocked by AMG487, which can be demonstrated by the decreased expression of IL-6 and TNF-α, suppressing the nuclear translocation of p65 and elevating tight junction protein expression. Importantly, these changes can be reversed by 3-MA. The data suggest that autophagy inhibition exerted reversal effects on the amelioration of CXCR3 inhibition in inflammation and the tight junction expression of LPS-treated IPEC-J2 cells.

## 5. Conclusions

In conclusion, this paper presents evidence that CXCR3 inhibition attenuates LPS-induced inflammation and intestinal barrier damage in CXCR3^−/−^ mice and IPEC-J2 cells by inducing autophagy. The results will provide instructions and a theoretical basis for future investigations by targeting CXCR3 in the treatment of intestinal injury (Figure 8).

## Figures and Tables

**Figure 1 cells-12-00182-f001:**
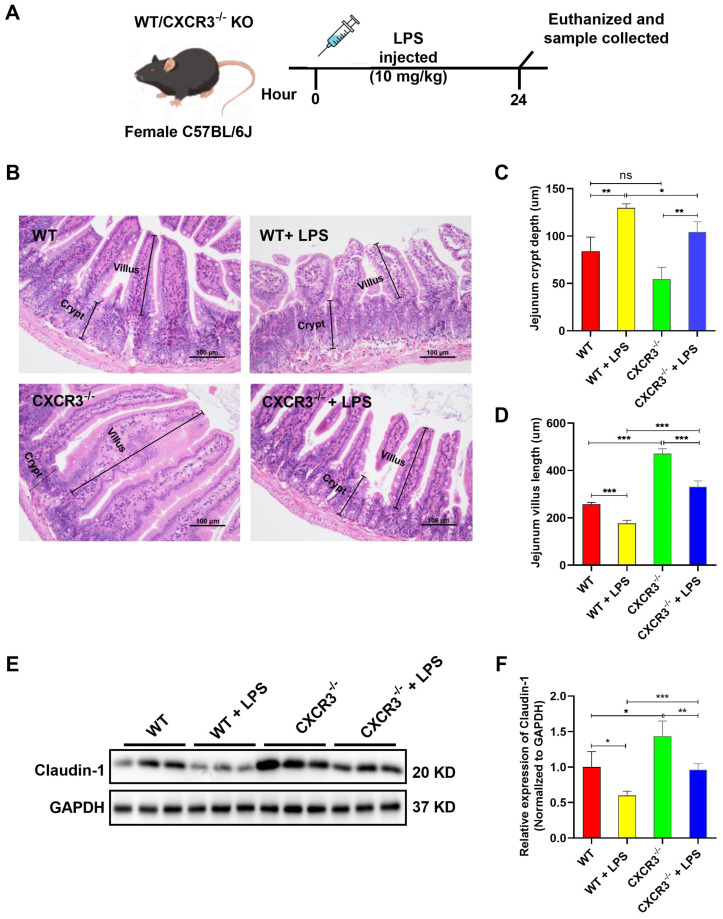
CXCR3 knockout attenuates intestinal epithelial structural damage and increases tight junction protein expression in the mouse intestine. C57BL/6J WT mice and CXCR3^−/−^ mice were intraperitoneally injected with LPS 10 mg/kg. After 24 h, the mice were sacrificed, and samples were collected. The jejunum tissue was fixed in formalin for HE staining and observed using an Olympus microscope. The protein was extracted for Western Blot. (**A**) Mice were intraperitoneally injected with LPS. (**B**) HE staining of jejunum tissue. (**C**) Intestinal villi height. (**D**) Crypt depth. (**E**) Claudin-1 protein expression. (**F**) Claudin-1 protein optical density value. The protein optical density value was analyzed by Image J 1.8.0 software (National Institutes of Health, Bethesda, MD, USA). Values represent the mean ± SEM. of three independent experiments. Significant differences are represented by * *p* < 0.05, ** *p* < 0.01, *** *p* < 0.001, and ns (not significant).

**Figure 2 cells-12-00182-f002:**
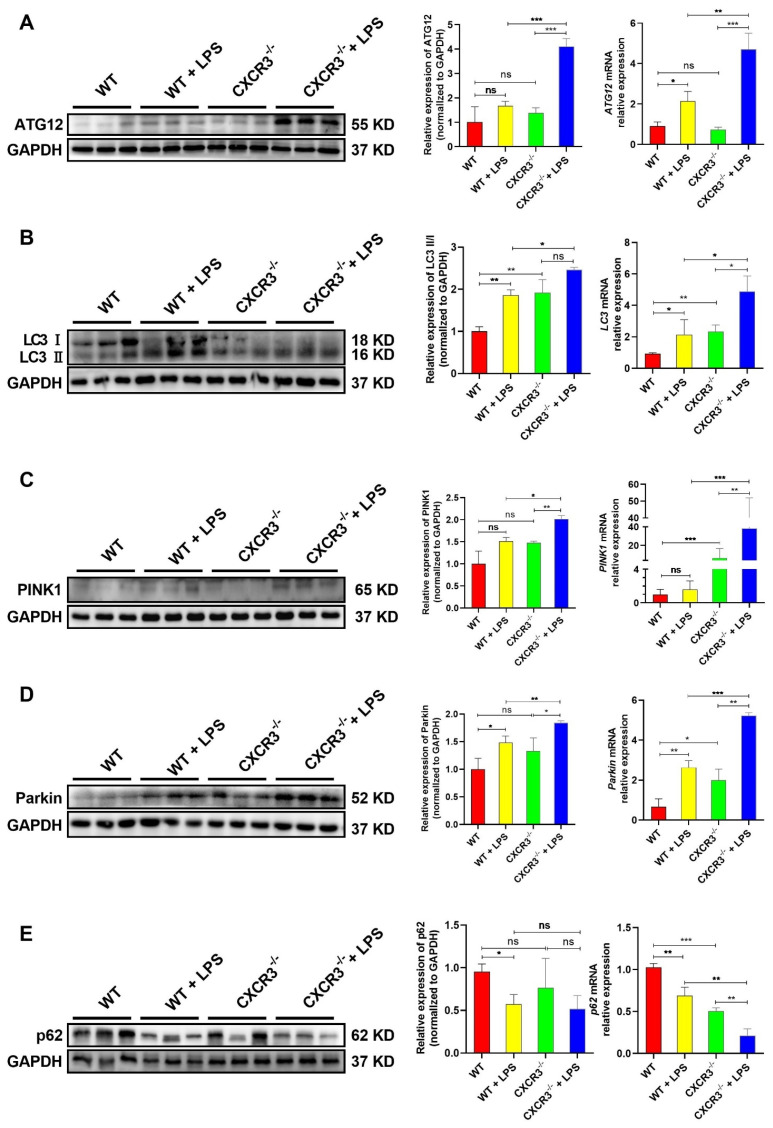
CXCR3 knockout promotes autophagy in the mouse intestine. C57BL/6J WT mice and CXCR3^−/−^ mice were intraperitoneally injected with LPS 10 mg/kg. After 24 h, the mice were sacrificed, and the intestine was collected. The mRNA and protein of jejunum tissue were extracted. qPCR and Western Blot assays were conducted to investigate the mRNA and protein expression of autophagy-related genes. (**A**) ATG12 mRNA and protein expression. (**B**) LC3 mRNA and protein expression. (**C**) PINK1 mRNA and protein expression. (**D**) Parkin mRNA and protein expression. (**E**) p62 mRNA and protein expression. The protein optical density values were analyzed by Image J software. Values represent the mean ± SEM. of three independent experiments. Significant differences are represented by * *p* < 0.05, ** *p* < 0.01, *** *p* < 0.001, and ns (not significant).

**Figure 3 cells-12-00182-f003:**
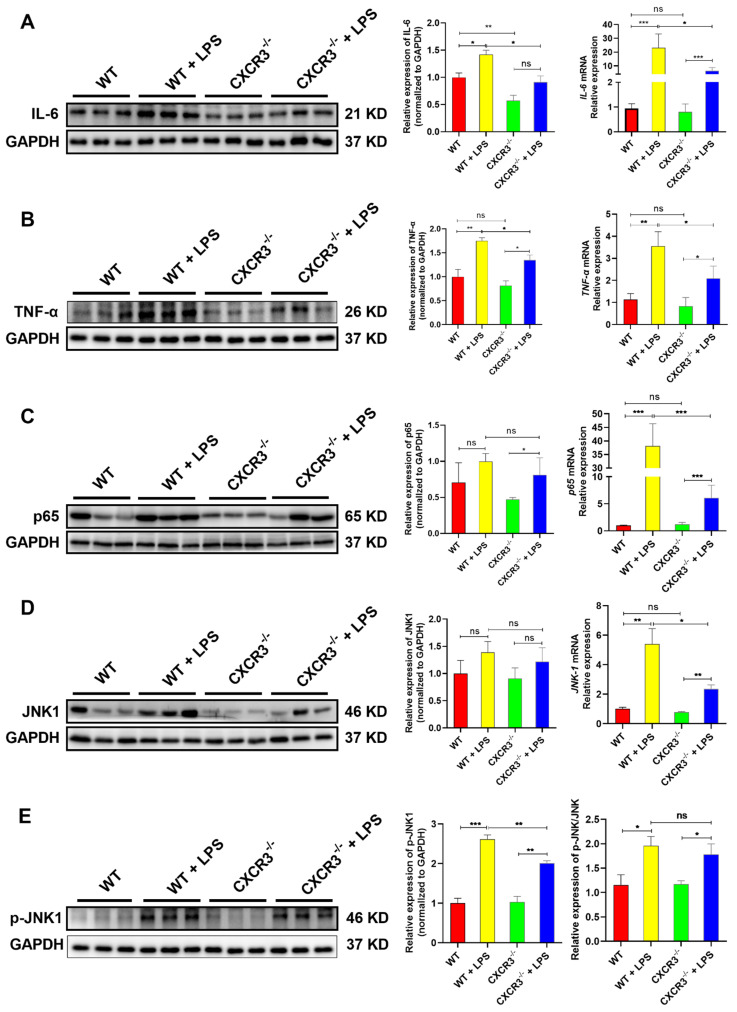
CXCR3 knockout inhibits the activation of the NF-κB signaling pathway in the mouse intestine. C57BL/6J WT mice and CXCR3^−/−^ mice were intraperitoneally injected with LPS 10 mg/kg. After 24 h, the mice were sacrificed. The mRNA and protein of jejunum tissue were extracted. qPCR and Western Blot assays were used to investigate the mRNA and protein expression of the NF-κB signaling pathway. (**A**) IL-6 mRNA and protein expression. (**B**) TNF-α mRNA and protein expression. (**C**) p65 mRNA and protein expression. (**D**) JNK1 mRNA and protein expression. (**E**) JNK1 phosphorylation. The protein optical density values were analyzed by Image J software. Values represent the mean ± SEM. of three independent experiments. Significant differences are represented by * *p* < 0.05, ** *p* < 0.01, *** *p* < 0.001, and ns (not significant).

**Figure 4 cells-12-00182-f004:**
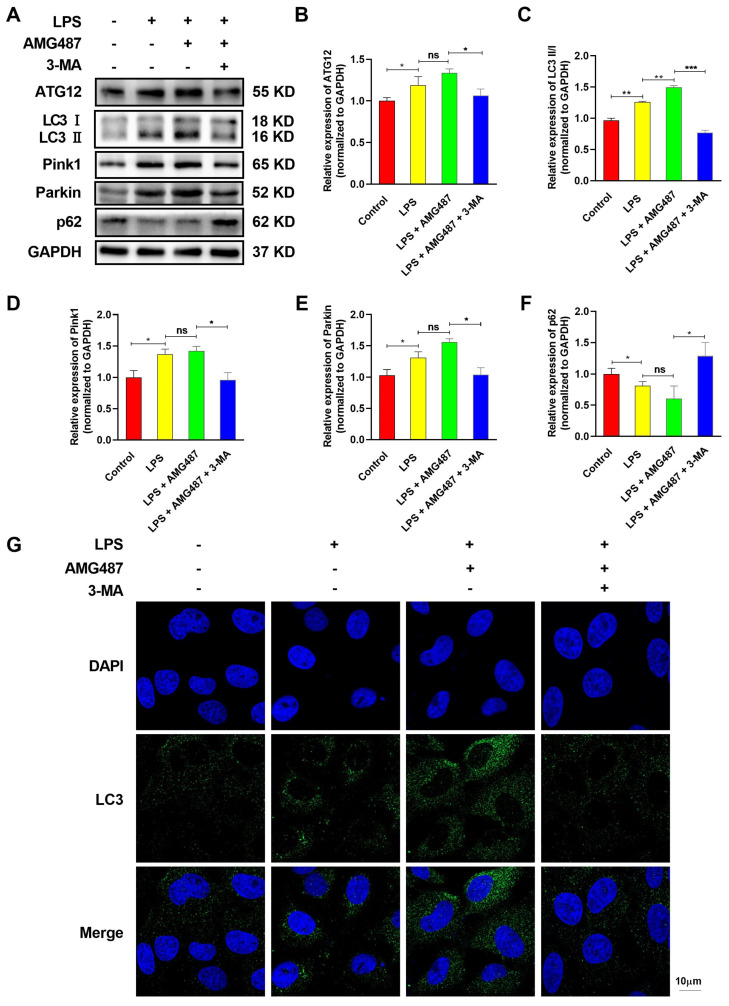
CXCR3 inhibition promotes autophagy in IPEC-J2 cells. The cells were divided into four groups, namely, the Control group, LPS (10 μg/mL) group, LPS + AMG487 (1 μM) group, and LPS + AMG487 + 3-MA (5 mM) group. The cells were treated for 12 h, and the protein was extracted for Western Blot. Indirect immunofluorescence assays were explored. (**A**) The expression of autophagy-related proteins in IPEC-J2 cells. (**A**,**B**) Parkin expression, (**A**,**C**) PINK1 expression, (**A**,**D**) p62 expression, (**A**,**E**) LC3 expression, and (**A**,**F**) ATG12 expression. (**G**) Indirect immunofluorescence laser confocal imaging of the distribution of LC3. The protein optical density values of autophagy-related genes were analyzed by Image J software. Values represent the mean ± SEM. of three independent experiments. Significant differences are represented by * *p* < 0.05, ** *p* < 0.01, *** *p* < 0.001, and ns (not significant).

**Figure 5 cells-12-00182-f005:**
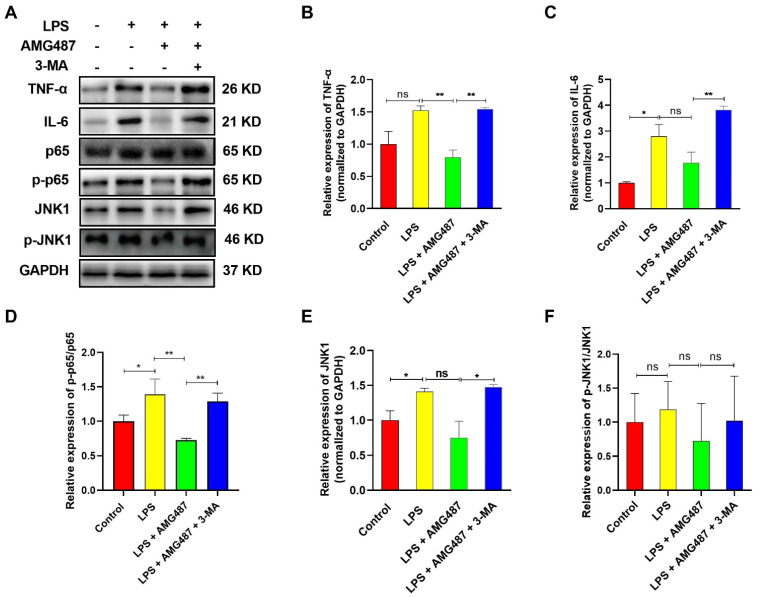
CXCR3 inhibition blocks NF-κB signaling pathway activation by promoting autophagy in the IPEC-J2 cells. The cells were divided into four groups, namely, the Control group, LPS (10 μg/mL) group, LPS + AMG487 (1 μM) group, and LPS + AMG487 + 3-MA (5 mM) group. The cells were treated for 12 h, and the protein was extracted for Western Blot. (**A**) The expression of NF-κB pathway-related proteins in IPEC-J2 cells. (**A**,**B**) TNF-α expression. (**A**,**C**) IL-6 expression. (**A**,**D**) p-p65/p65 expression. (**A**,**E**) JNK1 expression. (**A**,**F**) p-JNK1/JNK1 expression. Image J software analyzed the optical density values of NF-κB signaling pathway-related proteins. Values represent the mean ± SEM. of three independent experiments. Significant differences are represented by * *p* < 0.05, ** *p* < 0.01, and ns (not significant).

**Figure 6 cells-12-00182-f006:**
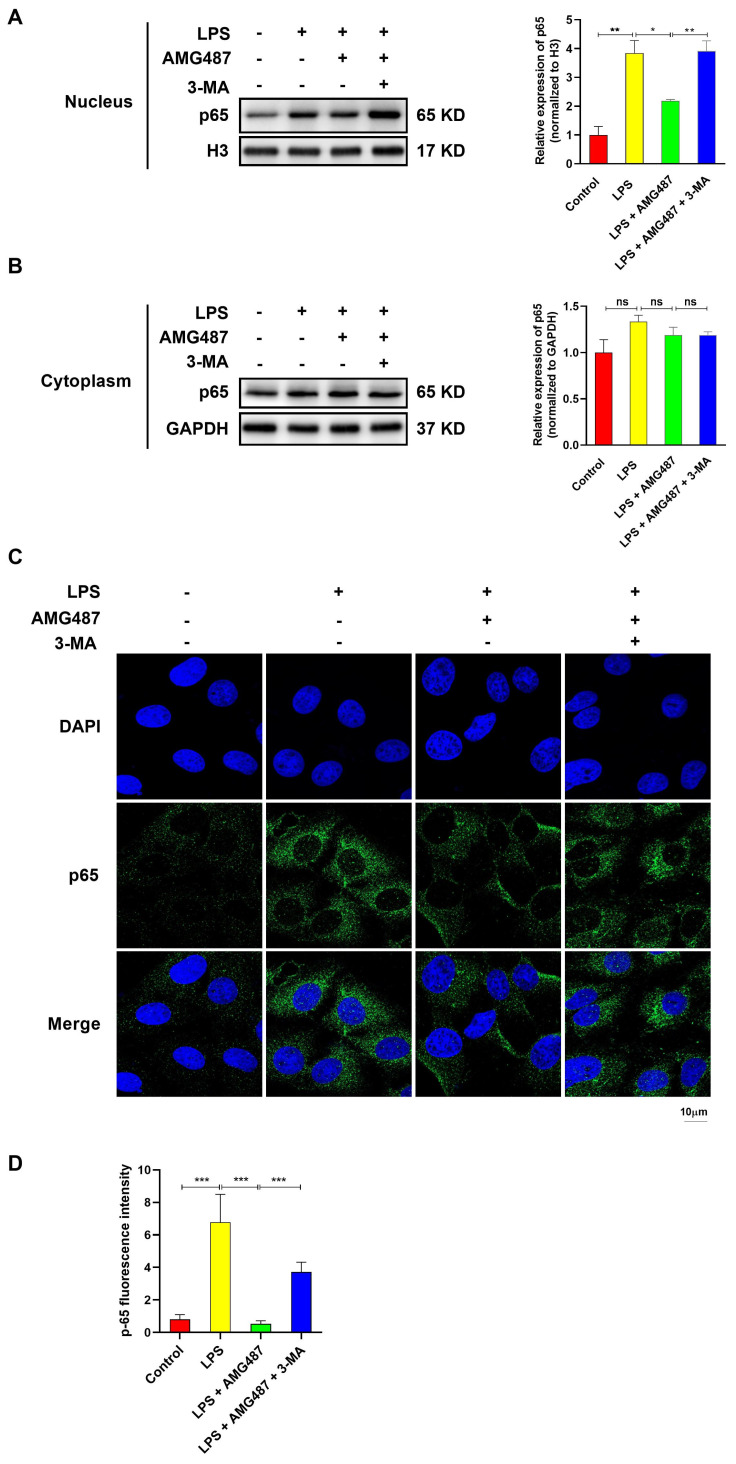
CXCR3 inhibition blocks p65 nuclear translocation by promoting autophagy in the IPEC-J2 cells. The cells were divided into four groups, namely, the Control group, LPS (10 μg/mL) group, LPS + AMG487 (1 μM) group, and LPS + AMG487 + 3-MA (5 mM) group. The cells were treated for 12 h. Nuclear and cytoplasmic proteins were extracted for Western Blot. An indirect immunofluorescence assay was explored. (**A**) The expression of p65 in the nucleus. (**B**) The expression of p65 in the nucleus cytoplasm. (**C**) Cell sub-localization of p65 with indirect immunofluorescence. (**D**) p-65 fluorescence intensity. The optical density values of p65 were detected by Image J software. Values represent the mean ± SEM. of three independent experiments. Significant differences are represented by * *p* < 0.05, ** *p* < 0.01, *** *p* < 0.01 and ns (not significant).

**Figure 7 cells-12-00182-f007:**
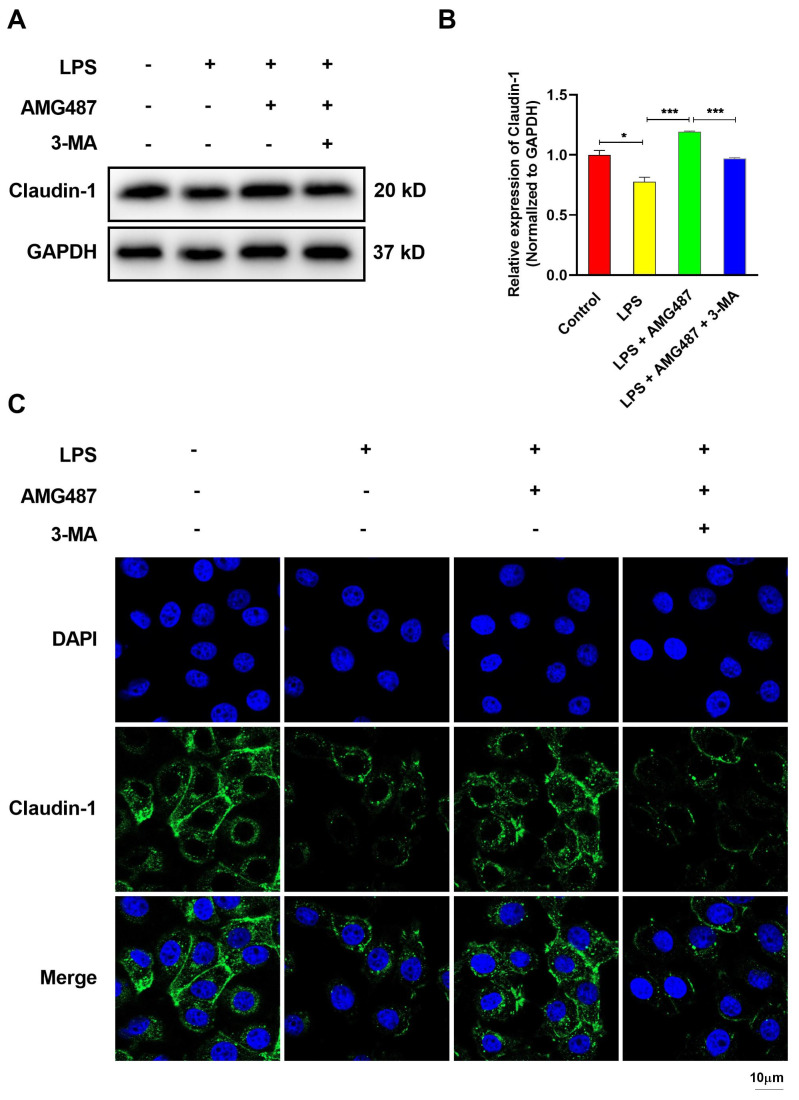
CXCR3 inhibition promotes tight junction protein expression by elevating autophagy in the IPEC-J2 cells. The cells were divided into four groups, namely, the Control group, LPS (10 μg/mL) group, LPS + AMG487 (1 μM) group, and LPS + AMG487 + 3-MA (5 mM) group. The cells were treated for 12 h, and the protein was extracted for Western Blot. Indirect immunofluorescence assays were explored. (**A**) Claudin-1 protein expression. (**B**) Image J software analysis of Claudin-1 protein optical density values. (**C**) Indirect immunofluorescence confocal imaging of the distribution of the Claudin-1 protein. Values represent the mean ± SEM. of three independent experiments. Significant differences are represented by * *p* < 0.05, *** *p* < 0.001, and ns (not significant).

**Figure 8 cells-12-00182-f008:**
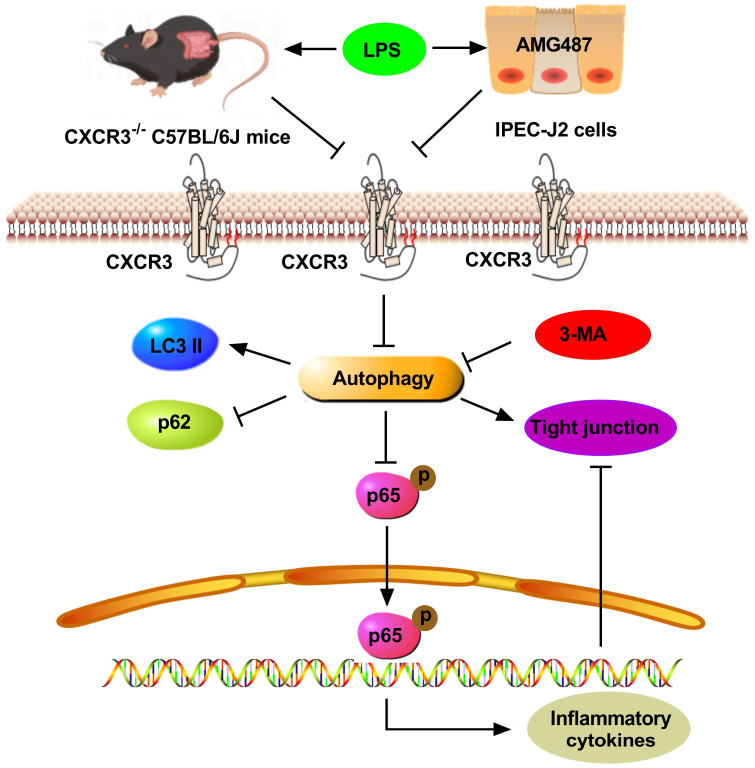
The schematic illustration of the underlying mechanism of CXCR3 inhibition-mediated autophagy alleviating inflammation and intestinal dysfunction. As a result, CXCR3 knockout and inhibition alleviate intestinal histological damage, increase tight junction expression, and restrain the NF-kB pathway in vitro and in vivo by autophagy elevation, which could be a therapeutic strategy by targeting CXCR3 against LPS-induced inflammation and intestinal dysfunction.

**Table 1 cells-12-00182-t001:** The primer sequence of qPCR.

Name	Sense (5′-3′)	Antisense (5′-3′)
*Pink1*	TGTCAGGAGATCCAGGCAATTTTTA	CTTCAGGACGACATCCGGGC
*Parkin*	ACAGAGACCACGGAGGAGAA	TAACTGGACCTCTGGCTGCT
*LC3*	TCCTGGATAAGACCAAGTTTCTG	ATAGATGTCAGCGATGGGTGTG
*p62*	ACGGAGTACCTGAACCCTCT	CTCGAGTCACAGTGGACCCT
*IL-6*	AACGATGATGCACTTGCAGA	TGGTACTCCAGAAGAAGACCAGAGG
*TNF-α*	GAGGCACTCCCCCAAAAGAT	CACTTGGTGGTTTGCTACGAC
*p65*	TGAACTTGTGGGGAAGGACTG	AGGTCTGTTTGGAAACTGGAGA
*JNK1*	TCTCCTTTAGCACAGGTGCAG	CTGCTGTCTGTATCCGAGGC
*ATG12*	CTTACGGATGTCTCCCCCAGA	ATGAGTCCTTGGATGGTTGG
*GAPDH*	AAATGGTGAAGGTCGGTGTGAAC	CAACAATCTCCACTTTGCCACTG

## Data Availability

Data will be made available on request.

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
