# Peer review of "CXCR3 Inhibition Blocks the NF-κB Signaling Pathway by Elevating Autophagy to Ameliorate Lipopolysaccharide-Induced Intestinal Dysfunction in Mice"

_cells, 2023, doi:10.3390/cells12010182_

Round 1
Reviewer 1 Report
This study showed that CXCR3 deteriorated LPS-induced intestinal inflammation by suppressing autophagy, by using CXCR3 knockout mice and CXCR3 inhibitor-treated IPEC-J2 cells. Although there were no particular problems with the results and conclusions, there are some points that must be confirmed as follows:
Major points
1. Line 165: Please indicate the analytical method used for the post-hoc analysis.
2. 2.5 Western Blot: Please describe the extraction method for nuclear and cytoplasmic proteins.
3. 2.6. Indirect immunofluorescence: Were the authors able to observe the fluorescence on a plastic plate instead of a glass dish?
4. Figure 1B-D: It would be helpful to indicate the location of the jejunal crypts and villus in the photograph.
5. The lysates extracted from the jejunum tissues of WT and CXCR3 mice should be used to confirm CXCR3 protein expression levels.
6. The authors should also check whether IPEC-J2 cells express CXCR3 protein. There is concern that the effect of AMG487 is artificial if it is not accompanied by CXCR3 expression.
7. Related to point 6, do IPEC-J2 cells secrete CXCL3 or other ligands for CXCR3?
8: Figure 5: JNK1 phosphorylation levels should also be analyzed.
9. Figure 6C: The fluorescence of nuclear p65 should be quantified by ImageJ, as it is not possible to determine whether p65 is translocated into the nucleus just by looking at the photograph.
10. Figure 8 is a little confusing. Especially, it is difficult how to interpret the vector between autophagy and LC3II and that between autophagy and p62. Furthermore, doesn’t CXCR3 suppress autophagy?
Minor points
1. The authors use the word “knockdown” a lot, but they should all be changed to “knockout”, because no experiments have been done with siRNA.
2. Line 329-331: The alphabetical indication of Figure 4 is incorrect.
3. Line 430-434: p65 phosphorylation levels were not analyzed in this experiment.
4. Line 442 and 447: Are “elevated” and “increase” both correct?
Author Response
- Line 165: Please indicate the analytical method used for the post-hoc analysis.
We thank the reviewer for pointing this issue.
The analytical method used for the post-hoc analysis was indicated in the revision.
Data are expressed as the mean ± SEM of at least three independent experiments for each experimental group. Statistical analysis was performed on the group by using One-way analysis of variance (ANOVA) followed by Tukey’s post hoc test in groups of more than two were used to establish statistical significance with GraphPad Prism version 9.0 (* P < 0.05; ** P < 0.01; *** P < 0.001; ns indicates no significance).
- 2.5 Western Blot: Please describe the extraction method for nuclear and cytoplasmic proteins.
We thank the reviewer for pointing this issue.
The extraction method for nuclear and cytoplasmic proteins has been described in the revision.
For nuclear protein preparations, cells were washed twice with PBS, and the proteins in the nucleus were extracted using a Nuclear and Cytoplasmic Protein Extraction Kit (Beyotime, Beijing, China) following the manufacturer’s protocol.
- 2.6. Indirect immunofluorescence: Were the authors able to observe the fluorescence on a plastic plate instead of a glass dish?
We thank the reviewer for pointing this issue.
The fluorescence was observed on glass dish with a fluorescence microscope or a confocal microscope.
- Figure 1B-D: It would be helpful to indicate the location of the jejunal crypts and villus in the photograph.
We thank the reviewer for pointing this issue.
The location of the jejunal crypts and villus in the photograph has been indicated in the revision.
- The lysates extracted from the jejunum tissues of WT and CXCR3 mice should be used to confirm CXCR3 protein expression levels.
We thank the reviewer for pointing this issue.
CXCR3 expression was detected in WT and CXCR3-/- mice, and the results showed that CXCR3-/- mice cannot express CXCR3 protein compared with the WT mice. The results were uploaded as Supplementary 1.
- The authors should also check whether IPEC-J2 cells express CXCR3 protein. There is concern that the effect of AMG487 is artificial if it is not accompanied by CXCR3 expression.
We thank the reviewer for pointing this issue.
IPEC-J2 cells are intestinal porcine enterocytes isolated from the jejunum of a neonatal unsuckled piglet. The IPEC-J2 cell line is unique as it is derived from the small intestine and is neither transformed nor tumorigenic in nature (The Impact of Food Bioactives on Health, 2015). In this study, CXCR3 expression in IPEC-J2 cells was confirmed by Western Blot. The result was uploaded as Supplementary 2.
- Related to point 6, do IPEC-J2 cells secrete CXCL3 or other ligands for CXCR3?
We thank the reviewer for pointing this issue.
We have confirmed that IPEC-J2 cells secrete CXCL3. Unfortunately, ligands for CXCR3 secreted by IPEC-J2 have not detected.
8: Figure 5: JNK1 phosphorylation levels should also be analyzed.
We thank the reviewer for pointing this issue.
JNK1 phosphorylation levels have been analyzed in the revision.
- Figure 6C: The fluorescence of nuclear p65 should be quantified by ImageJ, as it is not possible to determine whether p65 is translocated into the nucleus just by looking at the photograph.
We thank the reviewer for pointing this issue.
The fluorescence of nuclear p65 has been quantified by Image J in the revision (Figure 6D).
- Figure 8 is a little confusing. Especially, it is difficult how to interpret the vector between autophagy and LC3II and that between autophagy and p62. Furthermore, doesn’t CXCR3 suppress autophagy?
We thank the reviewer for pointing this issue.
Figure 8 has been revised in the manuscript.
Minor points
- The authors use the word “knockdown” a lot, but they should all be changed to “knockout”, because no experiments have been done with siRNA.
Thank the reviewer's comment.
In the revised manuscript, “knockdown” has been changed to “knockout”.
- Line 329-331: The alphabetical indication of Figure 4 is incorrect.
We thank the reviewer for pointing this issue.
The alphabetical indication of Figure 4 has been revised in the manuscript.
- Line 430-434: p65 phosphorylation levels were not analyzed in this experiment.
We thank the reviewer for pointing this issue.
Nuclear and cytoplasmic proteins were extracted following CXCR3 inhibition. p65 nuclear translocation was detected by Western Blot. The expression of p65 was mainly explored in nuclear and cytoplasmic in the manuscript.
- Line 442 and 447: Are “elevated” and “increase” both correct?
We thank the reviewer for pointing this issue.
“increase” should be “decrease”. We have revised it in the manuscript.
Reviewer 2 Report
Thank you for this good work
Author Response
Thanks for comments
Reviewer 3 Report
This paper studied the effects of CXCR3 on autophagy of intestinal epithelial cells in vivo and vitro, which provides a new theoretical basis and ideas for the prevention and treatment of intestinal inflammatory diseases by targeting CXCR3. There are some problems, which should be solved before it is considered for publication.
1- Language should be improved in this paper. Some sentences contain grammatical mistakes, such as, in the Abstract, “provide theoretical basis and ideas” would be “provide a theoretical basis and ideas” and “The results showed that knockdown of CXCR3…” would be “The results showed that the knockdown of CXCR3…”. Please check the manuscript carefully for relevant errors.
2- The introduction needs to add the relevant research background of the NF-κ B signaling pathway and other aspects.
3- Please check the description of the results. For example, Lin181, Figure 1 A-C should be Figure 1 B-D from the results.
4- Please make sure that the names of instruments and materials used are correct. For example, “Olmpus upright microscope” would be “Olympus upright microscope”, and “QPCR” would be “qPCR”. Line 334, the upper case and lower case should be consistent. Line 443, Immunofluorescence should be immunofluorescence.
5- The background on “autophagy” in 3.2 of the Result can be placed in the Introduction section.
6- The previous research bases of CXCR3 antagonist AMG487 were not discussed in the part of discussion and conclusion.
7- What is the relationship between autophagy and inflammation.
Author Response
- Language should be improved in this paper. Some sentences contain grammatical mistakes, such as, in the Abstract, “provide theoretical basis and ideas” would be “provide a theoretical basis and ideas” and “The results showed that knockdown of CXCR3…” would be “The results showed that the knockdown of CXCR3…”. Please check the manuscript carefully for relevant errors.
We thank the reviewer for pointing this issue.
We have modified language throughout the text as appropriate. The errors have been revised in the manuscript and marked with red.
- The introduction needs to add the relevant research background of the NF-κ B signaling pathway and other aspects.
We thank the reviewer for pointing this issue.
The relevant research background has been added in the introduction.
- Please check the description of the results. For example, Lin181, Figure 1 A-C should be Figure 1 B-D from the results.
We thank the reviewer for pointing this issue.
The description of the results has been checked and the errors has been revised in the manuscript.
- Please make sure that the names of instruments and materials used are correct. For example, “Olmpus upright microscope” would be “Olympus upright microscope”, and “QPCR” would be “qPCR”. Line 334, the upper case and lower case should be consistent. Line 443, Immunofluorescence should be immunofluorescence.
We thank the reviewer for pointing this issue.
The errors have been revised in the manuscript.
- The background on “autophagy” in 3.2 of the Result can be placed in the Introduction section.
We thank the reviewer for pointing this issue.
In this section, we mainly introduce the relationship between autophagy and immune response. It is necessary to discuss the background on “autophagy” in 3.2 of the Result.
- The previous research bases of CXCR3 antagonist AMG487 were not discussed in the part of discussion and conclusion.
We thank the reviewer for pointing this issue.
The function of AMG487 has been discussed in the discussion. Some references have been added in the revision.
- What is the relationship between autophagy and inflammation.
Response
We thank the reviewer for pointing this issue.
The relationship between autophagy and inflammation has been discussed in the manuscript.
Reviewer 4 Report
In this manuscript, the authors demonstrated that CXCR3 inhibition reversed LPS-induced intestinal barrier damage and alleviated NF-κB signaling pathway via enhancing autophagy by using in vivo mouse model and in vitro IPEC-J2 cells. They showed several interesting findings and I have a few comments:
In the Abstract. For the first time, the abbreviation shall be given the full name.
Why choose pig intestinal epithelial cells (IPEC-J2) as the cell model instead of mouse intestinal cells?
In vivo mouse experiment, why not detect the p-p65 expression?
The name of genes based on mRNA expression need to use italics to distinguish genes expressed in proteins.
The author needed to show the expression of CXCR3 after using inhibitor treatment to ensure gene expression is inhibited.
For the intestinal barrier gene, authors only choose the Claudin1, why?
Author Response
In the Abstract. For the first time, the abbreviation shall be given the full name.
Why choose pig intestinal epithelial cells (IPEC-J2) as the cell model instead of mouse intestinal cells?
We thank the reviewer for pointing this issue.
IPEC-J2 cells are intestinal porcine enterocytes isolated from the jejunum of a neonatal unsuckled piglet. The IPEC-J2 cell line is unique as it is derived from the small intestine and is neither transformed nor tumorigenic in nature (The Impact of Food Bioactives on Health, 2015). IPEC-J2 cell is wildly used in mouse model study (Food Funct, 2022,18;13(14):7507-7519; Food Res Int,2021,139:109840)
In vivo mouse experiment, why not detect the p-p65 expression?
We thank the reviewer for pointing this issue.
In the present study, only total p65 was detected in vivo mouse experiment. The results indicated that the changes show significant difference. Unfortunately, p-p65 expression has not detected.
The name of genes based on mRNA expression need to use italics to distinguish genes expressed in proteins.
We thank the reviewer for pointing this issue.
We have revised in the manuscript according to the comments.
The author needed to show the expression of CXCR3 after using inhibitor treatment to ensure gene expression is inhibited.
We thank the reviewer for pointing this issue.
We confirmed that the expression of CXCR3 is inhibited following AMG487 treatment.
For the intestinal barrier gene, authors only choose the Claudin1, why?
We thank the reviewer for pointing this issue.
The tight junction is a protein complex established by interactions between members of the claudin, zonula occludens, and tight junction-associated MARVEL protein (TAMP) families. Claudin 1, the integral tight junction proteins that regulate paracellular permeability and cell polarity, are frequently dysregulated in injury. In the present study, Claudin1 was chosen as a marker to reflect the intestinal barrier injury.
Round 2
Reviewer 1 Report
I am happy with the revisions made by the authors.